# The Pathogenic Role of Very Low Density Lipoprotein on Atrial Remodeling in the Metabolic Syndrome

**DOI:** 10.3390/ijms21030891

**Published:** 2020-01-30

**Authors:** Hsiang-Chun Lee, Yi-Hsiung Lin

**Affiliations:** 1Center for Lipid Biosciences, Kaohsiung Medical University Hospital, Kaohsiung 807, Taiwan; 2Lipid Science and Aging Research Center, Kaohsiung Medical University, Kaohsiung 807, Taiwan; caminolin@gmail.com; 3Division of Cardiology, Department of Internal Medicine, Kaohsiung Medical University Hospital, Kaohsiung Medical University, Kaohsiung 807, Taiwan; 4Department of Internal Medicine, School of Medicine, College of Medicine, Kaohsiung Medical University, Kaohsiung 807, Taiwan; 5Institute/Center of Medical Science and Technology, National Sun Yat-sen University, Kaohsiung 807, Taiwan

**Keywords:** very low density lipoprotein (VLDL), atrial remodeling, metabolic syndrome, atrial cardiomyopathy, atrial fibrillation, lipotoxicity

## Abstract

Atrial fibrillation (AF) is the most common persistent arrhythmia, and can lead to systemic thromboembolism and heart failure. Aging and metabolic syndrome (MetS) are major risks for AF. One of the most important manifestations of MetS is dyslipidemia, but its correlation with AF is ambiguous in clinical observational studies. Although there is a paradoxical relationship between fasting cholesterol and AF incidence, the benefit from lipid lowering therapy in reduction of AF is significant. Here, we reviewed the health burden from AF and MetS, the association between two disease entities, and the metabolism of triglyceride, which is elevated in MetS. We also reviewed scientific evidence for the mechanistic links between very low density lipoproteins (VLDL), which primarily carry circulatory triglyceride, to atrial cardiomyopathy and development of AF. The effects of VLDL to atria suggesting pathogenic to atrial cardiomyopathy and AF include excess lipid accumulation, direct cytotoxicity, abbreviated action potentials, disturbed calcium regulation, delayed conduction velocities, modulated gap junctions, and sarcomere protein derangements. The electrical remodeling and structural changes in concert promote development of atrial cardiomyopathy in MetS and ultimately lead to vulnerability to AF. As VLDL plays a major role in lipid metabolism after meals (rather than fasting state), further human studies that focus on the effects/correlation of postprandial lipids to atrial remodeling are required to determine whether VLDL-targeted therapy can reduce MetS-related AF. On the basis of our scientific evidence, we propose a pivotal role of VLDL in MetS-related atrial cardiomyopathy and vulnerability to AF.

## 1. Introduction

### 1.1. Atrial Fibrillation (AF) Remains a Major Health Problem, Especially for Aging Populations

AF is the most common sustained cardiac arrhythmia and it causes major morbidities such as systemic thromboembolism and heart failure, and it increases the mortality as well. Although there are many advances on cardiovascular therapies, such as medications and devices, AF-related burden to the healthcare systems and the worldwide AF prevalence are increasing in parallel [1,2,3,4]. For instance, the prevalence of AF in over 60-year-old individuals is estimated to increase from 3.9 million to 9 million by the year 2050 [5]. The scope of AF-related health problems has also been deliberated and pressed by the Heart Rhythm Society [6].

The causes for increasing incidence and thromboembolic complication of AF are suggested as being driven by two factors [3]. First, AF is more prevalent in the elderly population, and the population in most developed countries is aging [7]. On the other hand, age is also a major determinant for risk of stroke. For instance, individuals older than 75 years old carry an approximate fivefold increased risk for stroke than ones younger than 65 years old [8]. Second, the metabolic syndrome (MetS), which is another common health problem, also increases the risk for AF.

### 1.2. Atrial Remodeling and Atrial Cardiomyopathy Are Long-Term Changes Preceding the Occurrence of AF

With sufficient genetic predisposition, such as rare cases with a gap junction alpha-5 protein (GJA5) loss-of-function mutation, AF onset can occur at a relatively young age without coexisting structural changes [9]. In most common cases, AF develops upon a complex of genetic impacts and additional disease-related remodeling, and can be clinically manifested with electrical and/or structural remodeling, which is defined as atrial cardiomyopathy [10]. The following working definition of atrial cardiomyopathy has been proposed and published, and is based on experts’ consensus: “any complex of structural, architectural, contractile or electrophysiological changes affecting the atria with the potential to produce clinically-relevant manifestations” [10]. With this new definition, atrial cardiomyopathy can precede the onset of AF by a couple of decades [11]. 

During ventricular diastole, pressure is equalized between left ventricle and left atrial (LA). The LA remodeling with chamber dilatation can therefore develop in a condition of elevated filling pressure of LV, such as hypertension, myocardial infarction, heart failure or cardiomyopathy, and significant aortic valve diseases. LA enlargement has been shown an independent long-term predictor of cardiovascular events (hazard ratio, 2.0, *P* = 0.04) in the Pressioni Arteriose Monitorate E Loro Associazioni (PAMELA) cohort study [12] and also an incremental predictor of cardiovascular morbidity and mortality in type 2 diabetes mellitus patients [13]. P wave duration on electrocardiogram, which can reflect the electrical remodeling of atria, is shown to provide a simple risk marker for AF development in hypertensive patients [14]. 

In a clinical study performed on hypertensive patients receiving catheter ablation for arrhythmias, the abbreviated effective refractory periods in pulmonary veins, the prolonged interatrial and intra-atrial conduction times, and inducibility of AF are early electrical remodeling before the onset of AF [15]. Likewise, another study used cardiovascular magnetic resonance (CMR) imaging to investigate the structural and functional parameters of LA that can predict the risk of AF in 427 patients of hypertrophic cardiomyopathy [16]. Unlike AF-induced remodeling, in which the atria are affected selectively, most pathological processes that affect the atria also involve the ventricles to a greater or lesser extent [10]. 

### 1.3. MetS Is a Major Risk for Atrial Cardiomyopathy and AF

The MetS is a global phenomenon that may be caused by a complex of genetic predisposition, unhealthy diet, and sedentary lifestyle. The most commonly used definition for MetS is according to the guideline of the National Cholesterol Education Program Third Adult Treatment Panel (NCEP-ATP III) [17]. The diagnosis of MetS is determined on baseline examinations, when at least three of the following criteria are met: (1) central obesity, waist circumference ≥ 90 cm for men and ≥80 cm for women; (2) elevated triglycerides (≥150 mg/dL; 1.7 mmol/L); (3) low high-density lipoprotein cholesterol (HDL-C) level (<40 mg/dL; 1.03 mmol/L in men and <50 mg/dL; 1.29 mmol/L in women); (4) elevated blood pressure (systolic blood pressure ≥ 130 mmHg; diastolic blood pressure ≥ 85 mmHg); and (5) impaired glucose tolerance (fasting glucose ≥ 100 mg/dL; 5.6 mmol/L and/or a history of diabetes).

Previous studies on clinical observation have reported that MetS is associated with AF risk; this association is observed in western populations [18,19,20], and also in Asian countries [21,22,23]. MetS is common in AF patients without overt coronary heart diseases, and is independently correlated with increased risk of cardiac death [24]. In addition, MetS also has an impact on the outcome and recurrence risk of AF after a catheter ablation termination [25,26]. 

Among the MetS manifestations, hypertension and diabetes are strong independent risk for development of AF [27,28,29,30]. Aging, hypertension, and diabetes are also reported to be related to atrial fibrosis, which is also a risk of developing AF. Hypertension, in particular, is closely related to LA remodeling and AF, wherein approximately 30% of AF patients have hypertension. However, patients with MetS and hyperlipidemia often have co-existing hypertension and diabetes, and thus it is clinically difficult to identify the cause of atrial remodeling or dysfunction. Experimental results from rat hypertensive model showed that hypertension rapidly induces hypertrophy, fibrosis, and inflammation of LA, along with electrophysiological changes including atrial wavelength shortening and changes of Ca^2+^ current density [31], suggesting the pathogenic role of hypertension on atrial remodeling. In the case of hypertension clinically, the regression of electrocardiographic left ventricular hypertrophy (LVH) during antihypertensive therapy, regardless of the drug target, is associated with a lower likelihood of new-onset AF [32]. Likewise, the relationship between LVH and AF was reported in another study in which electrocardiographic LVH was shown a strong and independent predictor for recurrence of radiofrequency catheter ablation in patients with paroxysmal AF [33].

Diabetic cardiomyopathy or diabetes-induced cardiac dysfunction is a direct consequence of uncontrolled MetS. The clinical relevance of diabetes to progress of pathological cardiac remodeling, including electrical, structural, and autonomic remodeling in the atria, has been well recognized. However, the mechanisms underlying the increased susceptibility to AF in diabetes patients are incompletely understood. Some evidence has shown gap junction dysfunction and action potential alterations that may lead to atrial conduction abnormalities [34], and alteration in outward K^+^ channel expression [35], as electrical remodeling of the atrium.

Obesity is associated with the activation of the renin-angiotensin system, sympathetic nervous system, the release of adipokines, and ultimately lead to insulin resistance and hypertension. Obesity is associated with a 50% increase in AF risk and has also shown as an independent risk for AF [19,36]. Synergistically with hypertension, body weight status has been shown to be associated with new-onset AF [37]. 

Watanabe et al. reported that even in the absence of diabetes and hypertension, MetS is an independent risk factor for AF [38]. Zanoli et al. reported significant correlation between high-sensitive C-reactive protein (hsCRP) to atherosclerosis and arterial stiffness reflected by intima-medial thickness and aortic pulse wave velocity, respectively [39]. They suggested that MetS is characterized by an inflammation-dependent acceleration in cardiovascular ageing, which can lead to vascular dysfunction, and would contribute to make patients with MetS more susceptible to atrial remodeling [39]. 

### 1.4. The Ambiguous Relationship between Dyslipidemia and AF in Clinical Studies 

Among features in the MetS, dyslipidemia including hypertriglyceridemia and a low level of HDL-C, is clearly involved in atherosclerosis. Nevertheless, the real connection between dyslipidemia and AF remains unclear [3]. The relationship between dyslipidemia and AF is ambiguous among clinical studies. A paradox relationship between cholesterol and AF has been shown in the numbers of clinical studies [40,41,42] and also in Asian countries [43,44]. This paradox relationship indicates that low cholesterol levels are associated with higher risks for AF. The inverse correlation between plasma low density lipoprotein cholesterol (LDL-C) and total cholesterol to new onset AF during hospitalization is also observed in patients with acute ST-segment elevation myocardial infarction [45]. Similarly, total cholesterol and LDL-C levels are inversely correlated with AF recurrence after radiofrequency catheter ablation in women [46]. The causes of this paradoxical correlation between dyslipidemia and AF remain undetermined. Likewise, very low density lipoproteins (VLDL), which are derived by the cholesterol-poor small LDL, have been shown to be inversely associated with AF events [30]. 

Although plenty of clinical evidence exists that supports the paradoxical correlation between hypercholesterolemia and AF risk, on the other hand, the pleiotropic effects of statins have been shown to prevent AF development in numbers of clinical observational studies [47,48,49]. In the Framingham Heart Study, which had enrolled multi-ethnic non-AF men and women and followed up for a mean duration of 9.6 years, high triglyceride was found to be associated with a higher risk of AF [50]. These observations suggest that the inverse correlation between cholesterol and AF cannot exclude the contributing role of lipids on the pathogenesis in atrial remodeling and AF. 

Kim et al. reported that serum lipoproteins are different in middle-age, paroxysmal AF patients and ones without any AF. The differences include higher triglyceride content and higher oxidized lipid species in VLDL, lesser cholesterol content in LDL, and less anti-oxidant activities of HDL [51]. Alteration in characteristics of lipoproteins, including VLDL, and LDL from AF patients, have been shown to exhibit an easier uptake into macrophages by phagocytosis [51]. HDL from AF patients, on the other hand, exhibits decreased antioxidant activities and approximate 30% lower apolipoprotein AI (ApoA-I) expression compared to non-AF controls [52], also suggesting that the uptake of lipids may be much more important than the circulatory level for the impact of lipids on atrial remodeling. 

### 1.5. The Metabolism of VLDL

The plasma VLDL-TG concentration is regulated by a complex of multiple metabolic factors, such as insulin, adipose tissue, and intrahepatic triglycerides, and is also affected by race and sex [53,54]. Overproduction and/or delayed clearance of VLDL is considered as causing hypertriglyceridemia. The metabolism of VLDLs involves numbers of apolipoproteins. Apolipoprotein (apo) C-III is a glycoprotein that is synthesized by the liver and the intestine, and it is exchangeable between triglyceride-rich lipoproteins and HDL. VLDL-ApoC-III is shown overproduced with elevated concentration and causes delayed catabolism of triglycerides and ApoB in VLDLs [55]. On the other hand, apolipoprotein AII (Apo-AII), which is mostly associated with HDLs and related to insulin resistance, is a regulator of VLDL metabolism [56]. 

After meals, VLDL particles have been shown to be strongly bound and internalized into cells expressing the VLDL receptor (VLDLR) [57], such as cardiomyocytes. The VLDLR, a 118 kDa protein, is a member of LDL receptor family and it functions as a route for triglyceride-rich lipoproteins and remnants to be delivered into heart tissue, adipose tissue, endothelial cells, and the brain. In addition to lipoproteins, VLDLR can bind lipoprotein(a) (Lp(a)), receptor-associated protein, reelin, thrombospondin-1, proprotein convertase subtilisin/kexin type 9 (PCSK9), urokinase plasminogen activator inhibitor-1 complex, and tissue factor pathway inhibitor [57]. In endothelial cells, VLDLR interacts with fibrin, and as a result, transcellular leukocyte migration and inflammation are promoted [58]. 

The VLDL secretion of the liver is a normal chronical response to elevated glucose and insulin after meals. Following secretion from liver into the circulation, the size and contents of VLDLs change by interacting with several enzymes, such as phospholipid transfer protein (PLTP), cholesteryl ester transfer protein (CETP), and lipoprotein lipase (LPL). Hepatic PLTP modulates VLDL secretion from liver. In circulation, PLTP facilitates transfer of phospholipids from VLDLs to HDLs. CETP is expressed predominantly in the liver, spleen, and adipose tissue, and facilitates the exchange of cholesteryl esters from HDLs to TG-rich lipoproteins. Increased risk of AF for post-menopausal women has been shown associated with CETP B2B2/AA genotype [59]. This clinical observation is another piece of evidence suggesting the association of AF with an abnormal VLDL-related lipid metabolism. 

## 2. Cardiac Lipotoxicity

The heart is capable of utilizing a variety of substrates for energy sources, including glucose, fatty acids, lactate, and ketones. Under normal aerobic condition, the majority (approximate 70%) of ATP production is from utilization of lipids [60]. Responding to stress, the ATP production from lipid utilization is reduced. For instance, under pressure overload, the hypertrophic left ventricle utilizes more glucose and less fatty acids for ATP production [61]. 

The heart expresses LPL gene abundantly and the synthesized LPL from cardiomyocytes can be subsequently transferred to the luminal surface of endothelial cells by binding to the molecule heparan sulphate proteoglycan. On the other hand, LPL located over the cellular surface of cardiomyocytes also mediates cardiac triglyceride uptake [60,62]. Besides LPL-mediated hydrolysis, there is receptor-mediated endocytosis, including at least VLDLR and apolipoprotein B-48 (Apo-B48) receptor [63]. 

A relationship between myocardial VLDLR expression and circulatory VLDL has been suggested by studies since the VLDLR gene was cloned in 1992 [64,65]. Interestingly, interaction between VLDLR and LPL exists as well. Upon binding to the VLDLR, LPL hydrolysis was facilitated [66]. On the other hand, the receptor-associated protein that inhibits VLDLR also inhibits LPL activity [67]. Although it is well known that lipid-loaded lipoproteins or lipoprotein remnants enter cardiomyocytes via lipoprotein receptors, the physiological role of these receptors is still unclear [61]. For instance, the VLDLR is a member of the LDL receptor superfamily and binds ApoE-TG-rich VLDLs and chylomicrons [64]. VLDLRs have been shown to be abundantly expressed in the heart [61,64] and endothelial cells [68]. In a mouse model of myocardial infarction, the VLDLR was shown to promote lipotoxicity and increase mortality [69]. It is possible that VLDL-induced reduction of glucose transport into cardiomyocytes may impair metabolic stress-stimulated adaptation during ischemic injury [70]. 

Fine coupling between lipid uptake and its oxidation prevents excess accumulation of lipids in the cardiomyocytes. The excess lipid accumulation can cause structural damage, apoptosis, mitochondrial dysfunction, and increased fibrosis, and can lead to reduced cardiac function [71,72,73]. Several conditions, such as insulin resistance and type 2 diabetes mellitus, cause cardiac lipid accumulation [74,75]. Fast heart rate per se can induce lipid accumulation in LA appendage of a canine AF model, which is created by atrial rapid pacing. Metabolic impairments, such as alterations in the mitochondrial ATP synthase, decreased atrial ATP content, elevation of hypoxia-inducible factor 1α (HIF1α), and reduced LA capillary density, have been shown in the atria of chronic AF [76,77]. Marfella et al. also reported evidence of myocardial lipid accumulation in aortic stenosis patients with MetS and pressure-overloaded hearts [78]. 

Early in 1978, Nestel and Poyser reported significant, positive correlations between plasma VLDL cholesterol levels and cholesterol content of atrial tissues that are obtained at the time of coronary artery surgery [79]. Although they failed to find any correlation between tissue cholesterol and circulatory levels of LDL or HDL, this study remains an in vivo piece of evidence supporting the attributing role of VLDL to the magnitude of cholesterol influx into atrial tissues. 

## 3. VLDL of MetS Exhibits Cytotoxicity to Atrial Myocytes

From evaluating individuals with and without MetS, our colleagues reported the diversity of VLDLs, in which VLDL of MetS carries a more negative charge and is significantly cytotoxic to human vascular endothelial cells when compared with VLDL of non-MetS subjects [80]. Later we showed that VLDL of MetS patients induced cytosolic oxidative stress and exhibited more cytotoxicity to atrial myocytes (HL-1 cells) compared with VLDL of non-MetS subjects [81]. In vivo, intravenous administration of VLDL of MetS with 6 weeks’ duration induced myocytes apoptosis, which was demonstrated by histological study with in situ terminal deoxynucleotidyl transferase (TUNEL) staining. This finding was accompanied by increased lipid accumulation in atrial tissues (Figure 1) [81]. According to the latest expert consensus, excessive lipid accumulation is classified as EHRAS (for EHRA/HRS/APHRS/SOLAECE) Pathology Class IVf of atrial cardiomyopathy [10].

## 4. VLDL of MetS Induces Atrial Remodeling, Ventricular Hypertrophy, and Vulnerability to AF 

In our VLDLs injection mice, LVH developed after 4 to 6 weeks (Figure 1). Nevertheless, only the MetS group had left atrial dilatation. The atrial dilatation takes place before ventricular hypertrophy [81], therefore the VLDL-induced atrial dilatation is considered as a primary change rather than secondary to the ventricular hypertrophy. Most pathological processes contributing to atrial remodeling also affect the ventricles; likewise, VLDL also induces ventricular hypertrophy. 

In addition to the cardiac remodeling, VLDL also causes ventricular function impairment. In our first publication on using the VLDL injection mouse model, there was a borderline reduced ejection fraction of the left ventricle in the MetS group of young mice [81]. In our second publication regarding in vivo effects of VLDL on cardiac function, the MetS-VLDL injection caused reduced contractility of ventricle of middle-aged mice [82]. 

In electrocardiography, young mice of the MetS group were found to have ventricular premature beats at baseline without any stress, and they also had AF after receiving an intraperitoneal injection of isoproterenol, a rapid-onset β-receptor agonist to mimic elevated sympathetic tone under stress [81]. We also observed spontaneous, unprovoked AF among half of the middle-aged mice receiving MetS-VLDL injection [81]. These findings provide direct evidence that VLDL of MetS can induce atrial remodeling and vulnerability to AF. According to the aforementioned results, we suggest that the atrial cardiomyopathy in MetS is related to VLDL-induced lipotoxicity. 

## 5. VLDL of MetS Induces Intra-Cardiac Conduction Delay via Modulation of Cardiac Gap Junction Cx40/43

The intra-cardiac conduction can be reflected by utilizing each component of electrocardiographic waves, including P waves, PR intervals, QRS durations, and QT intervals that indicate the atrial depolarization/repolarization, impulse conduction from the atria to the AV node, ventricular depolarization, and finish of ventricular repolarization, respectively. In the VLDL injection mice, the MetS group had significantly prolonged P waves, PR intervals, QRS durations, and QT intervals, indicating that the delayed cardiac conduction is intra-atrial, between atria-ventricular and intra-ventricular [82]. By using the optical mapping technique on ex vivo hearts, the slowed conduction velocities over bilateral atria and bilateral ventricles were demonstrated in the MetS group compared with the control and non-MetS VLDL group [82]. One of the possible causes of the cardiac conduction delay, intra-cardiac fibrosis, was excluded by our negative finding of the histological study. 

Gap junctions facilitate efficient electrical propagation in the heart and determine the intracardiac conduction throughout the atria, ventricles, and conduction system. We checked the expression levels of gap junction proteins Cx40 and Cx43, and their genes GJA5 and GJA1, respectively. Cx43 is expressed throughout the heart, whereas Cx40 is exclusively expressed in the atria, the His bundle, and Purkinje fibers. Our results found that the expression for both Cx40 (GJA5) and Cx43 (GJA1) was significantly reduced in the MetS mice atria. Besides down-regulation of gap junction proteins and gene expression, MetS-VLDL was found to also modulate the Cx40 and Cx43 proteins with enhanced *O*-GlcNAylation, which is postulated to affect the gap junction assembly and stability by interfering with the electrostatic balance on neighbor molecule interaction [82]. The *O*-GlcNAylation of Cx40 under high glucose condition has been reported to affect gap junction in endothelial cells [83]. Our report was the first to demonstrate the VLDL-induced *O*-GlcNAylation of Cx40 and Cx43 in atrial myocyte HL-1 cells. The possibility of gap junction dysfunction or altered gating property upon *O*-GlcNAcylation requires further investigation. 

## 6. VLDL of MetS Suppresses the Store-Operated Calcium Entry (SOCE) and Downstream Calcineurin-Nuclear Factor of Activated T Cells (NFAT) Pathway 

The cardiac hypertrophy is an adaptive capacity to stress and it is regulated by the calcineurin–nuclear factor of activated T cells (NFAT) pathway, which is linked to the stromal interaction molecule 1 (STIM1)/calcium release-activated calcium channel protein 1 (Orai1)-mediated store-operated Ca^2+^ entry (SOCE) [84,85]. We reported evidence that VLDL of MetS attenuates SOCE and down-regulates the calcineurin-NFAT pathways [86]. The STIM1 is a Ca^2+^-binding protein and the Ca^2+^ sensor in the sarcoplasmic reticulum (SR), and it interacts with Ca^2+^ inward channels on the plasma membrane to replenish the SR calcium storage [87]. On treatment with VLDL of MetS, STIM1 is down-regulated on its protein and mRNA expressions, and it is also modulated by *O*-GlcNAylation. These findings are associated the inhibition of the calcineurin-NFAT pathway [86], which has been shown to be critical in both pathological hypertrophy and cardiac adaptation to biomechanical stress. We suggest that in MetS, VLDL affects the cardiac adaptation capacity of atrium. 

## 7. VLDL of MetS Disturbs Myofilament Regulation

Sarcomere proteins of cardiomyocytes are regulated by a complex of mechanisms, and the alteration affects the motor function in which the interaction between thin and thick filaments leads to contraction of cardiomyocytes [88]. Genetic and hereditary cardiomyopathies, aging hearts, and heart failure are associated with predominant sarcomere proteins disorders [88,89]. On HL-1 cells treated with VLDL of MetS and on MetS group mice, alteration of sarcomere protein expression has been observed, including cMYBPC, desmin, troponin-T, troponin-I, and MLC2. This finding is associated with ultrastructural changes that are demonstrated by transmission electron microscopy, including disorganized Z-lines and sarcomeres, and morphological changes of mitochondria along the myofilaments [86]. We suggest that in atrial cardiomyopathy of MetS, VLDL causes the sarcomere derangement, which partially contributes to the impaired contractility in our VLDL-injected mice. 

## 8. The Pathogenic Role of VLDL in Atrial Cardiomyopathy and Future Directions 

On the basis of the available research findings thus far, as well as existing knowledge, we propose the pivotal and pathogenic role of VLDL in MetS-related atrial cardiomyopathy, manifested with both electrical and structural remodeling. The electrical remodeling includes triggered activities, delayed intra-cardiac conduction, vulnerability to AF upon sympathetic stimulation, and unprovoked AF. The structural remodeling includes atrial dilatation, ventricular dilatation, and hypertrophy associated with impaired contractility, along with increased myocardial apoptosis, excessive lipid accumulation, sarcomere disorders, and myofilament derangements, and also mitochondria morphological changes (Figure 2) [81,82,86]. 

Hypertriglyceridemia is a feature of MetS, and TG-rich chylomicrons and VLDL are largely produced and secreted after meals. Nakajima et al. reported that postprandial VLDL remnants have a higher affinity to the VLDLR compared to fasting VLDL, and they suggest both remnant lipoprotein-ApoB48 and -ApoB100 are endogenous VLDL remnants produced in the liver postprandially [57,91]. To understand the role of VLDL in atrial cardiomyopathy for human subjects, it is important to clarify postprandial concentrations and characteristics of VLDL and VLDL remnants in terms of the composition of apolipoproteins, lipid cores, charge, affinity to receptors, and favorability of internalization to cardiomyocytes. 

We hypothesize that VLDLR plays a determinant role in atrial cardiomyopathy of MetS. We examined the cardiac remodeling on a mouse model that carries VLDLR loss-of-function mutant [92] and will utilize cardiac-specific VLDLR gene knock-out mice to perform more experiments. The results will offer direct evidence for the role of VLDLR in the pathogenesis of MetS-related atrial cardiomyopathy. In addition to intra-cardiac effects, our colleagues have reported that VLDL isolated from subjects with MetS induces microglia activation associated with elevation of tumor necrosis factor (TNF-α) and prostaglandin E2 (PGE2). This finding indicates that VLDL in MetS can contribute to neuroinflammation [93]. The autonomic nervous system plays an important role in arrhythmogenesis, and we hypothesize that VLDL may affect the autonomic innervation or activities on the heart, and therefore result in vulnerability to arrhythmias, especially for individuals with MetS. On the other hand, the intestinal postprandial production of TG-rich chylomicron and the secretion of VLDL from the liver that may be feedbacked by the circulatory TG-rich lipoproteins/remnants, may be intervened by the autonomic function as well [94]. If this is true, the post-prandial VLDL might be the most important factor determining the pathogenesis and progress of MetS-related atrial cardiomyopathy and ultimately vulnerability to the development of AF, as well as AF-related cognitive dysfunction. Further human studies are required in terms of focusing on the effects/correlation of postprandial lipids on atrial remodeling and its progression to determine whether VLDL-targeted therapy can reduce the occurrence of MetS-related AF.

## Figures and Tables

**Figure 1 ijms-21-00891-f001:**
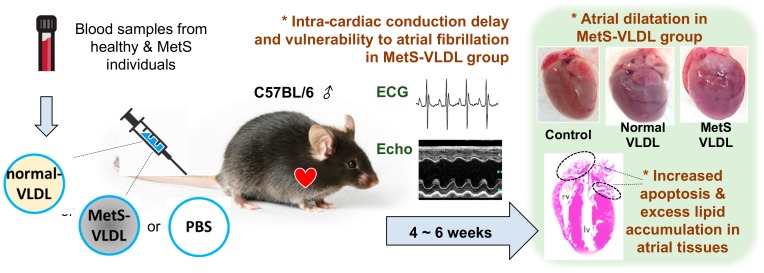
In vivo effects of very low density lipoproteins (VLDLs) on atrial remodeling of C57BL/6 male mice. Blood samples were collected from heathy volunteers and individuals with metabolic syndrome (MetS). Normal-VLDL and MetS-VLDL were isolated respectively. The VLDLs were injected via tail vein of the mouse three times a week for 6 consecutive weeks. The electrocardiography shows intra-cardiac conduction delay with significantly prolonged P waves and PR intervals in the MetS group. Upon isoproterenol challenge, some mice of the MetS group had atrial fibrillation whereas other groups did not. The echocardiography shows dilatation of left atria in the MetS group. In histological studies, MetS group were found to have increased lipid accumulation and increased myocyte apoptosis when compared with the control and normal-VLDL group.

**Figure 2 ijms-21-00891-f002:**
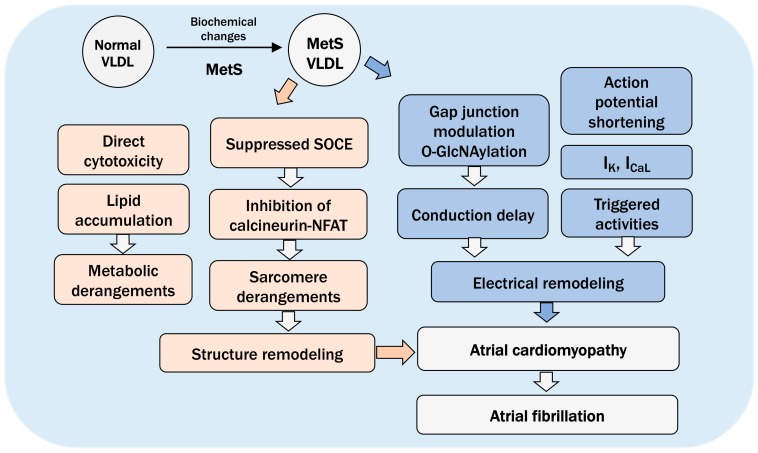
The pathogenic role of VLDL in MetS-related atrial cardiomyopathy, exhibited with structurally (in pink blocks) and electrical (in blue blocks) remodeling. The structural remodeling includes atrial dilatation, ventricular dilatation, and hypertrophy associated with impaired contractility, along with increased myocardial apoptosis, excessive lipid accumulation (consistent with EHRAS class IVf of atrial cardiomyopathy [10]), sarcomere disorders and myofilament derangements, and mitochondria morphological changes. The electrical remodeling includes shortening of action potentials [90], triggered activities, delayed intra-cardiac conduction, vulnerability to AF upon sympathetic stimulation, and unprovoked occurrence of AF. SOCE: store-operated calcium entry, NFAT: nuclear factor of activated T cells.

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
