# Peer review of "The Pathogenic Role of Very Low Density Lipoprotein on Atrial Remodeling in the Metabolic Syndrome"

_ijms, 2020, doi:10.3390/ijms21030891_

Round 1

Reviewer 1 Report

The author's described review from an atrial histological viewpoint is interesting. However, there is a slight bias in the review, and I feel it is necessary to revise the review to include a clinical perspective.

Although hyperlipidemia (VLDL) may play a part of role in the onset of LA remodeling and AF, what is generally interpreted is, triggered by myocardial stretch of LA, the RAA system in atrial tissue is activated and the fibrosis of LA is promoted (structural remodeling). This, in turn, causes myocardial conduction non-uniformity, giving the anatomical base on which AF is maintained. Once AF develops, its conduction delays over time and finally becomes persistent AF (electrical remodeling).

Therefore, I think that the following points need to be added.

Atrial fibrosis has attracted attention as a cause of AF. It has been reported that atrial fibrosis progresses with aging, hypertension (HT), diabetes (DM), etc., and becomes a risk of developing AF. HT is closely related to LA remodeling and AF (about 30% of AF patients are patients with HT). In addition, in patients with MetS and hyperlipidemia often have HT and DM at the same time, thus clinically, it is often difficult to identify the cause of atrial dysfunction. Therefore, careful discussion is required. What is the definition of atrial cardiomyopathy in the schematic diagram of Fig. 1? Does it mean a decrease in LA contractility? Active contraction (booster pump function) of LA disappears when AF develops.

It is appropriate to understand clinically from the viewpoint that electrical remodeling and structural remodeling occur in parallel, and that AF is fixed and eventually develops into atrial cardiomyopathy.

   Therefore, Atrial cardiomyopathy→Atrial fibrillation is skeptical. 

There are some cases showing AF reverse remodeling after ablation therapy in AF patients. How does author explain this phenomenon from viewpoint of MetS-related cardiomyopathy histologically? Is the pivotal role in line 33, the most important in line 322, etc. overemphasized?

Reviewer 2 Report

This review by Lee & Lin focused on the role of VLDL in the pathogenesis of atrial remodeling in the metabolic syndrome. As to the subject focused in the title, the review looks well balanced and provides the reader with some important findings underlying the role of lipid metabolism linking atrial remodeling with metabolic syndrome. However, I would suggest to bring to the fore the relationship between left ventricular hypertrophy, that is a major determinant of atrial fibrillation, and metabolic syndrome, as recently emphasized by Zanoli et al (Nutr Metab Cardiovasc Dis 2019). A vicious circle including insulin resistance, inflammation, arterial remodeling and vascular dysfunction, eccentric left ventricular hypertrophy and increased elastance, would contribute, in parallel with a derangement in lipid metabolism, to make patients with metabolic syndrome  more susceptible to atrial remodeling. In other words, the metabolic/inflammatory factors  and hemodynamic variables would work synergistically, and this concept should not be omitted. 

Author Response

Reviewer 2.
Comments and Suggestions for Authors
This review by Lee & Lin focused on the role of VLDL in the pathogenesis of atrial remodeling in the metabolic syndrome. As to the subject focused in the title, the review looks well balanced and provides the reader with some important findings underlying the role of lipid metabolism linking atrial remodeling with metabolic syndrome. However, I would suggest to bring to the fore the relationship between left ventricular hypertrophy, that is a major determinant of atrial fibrillation, and metabolic syndrome, as recently emphasized by Zanoli et al (Nutr Metab Cardiovasc Dis 2019).
A vicious circle including insulin resistance, inflammation, arterial remodeling and vascular dysfunction, eccentric left ventricular hypertrophy and increased elastance, would contribute, in parallel with a derangement in lipid metabolism, to make patients with metabolic syndrome more susceptible to atrial remodeling. In other words, the metabolic/inflammatory factors and hemodynamic variables would work synergistically, and this concept should not be omitted.
Response:
We would like to thank the review for pointing out the significance of the work and for the expert and very helpful comments. In the following we respond to these comments.
We had added statements regarding the relationship between left ventricular hypertrophy with atrial fibrillation (line 101 to 105; 230, 234-235) and statements regarding the inflammation to atrial remodeling to Section 1.3 (line 111 to 116). The Zanoli et al paper had been added to the reference
list.

Round 2

Reviewer 1 Report

Author has not responded to my each comments. For each reviewer's point, the author generally responds to the bulleted list first, and then corrects the text.

Author Response

---------------------------------------------------------

Specific Responses to the Reviewer Comments

---------------------------------------------------------
ijms-680843

Reviewer 1.

Comments and Suggestions for Authors

The author's described review from an atrial histological viewpoint is interesting. However, there is a slight bias in the review, and I feel it is necessary to revise the review to include a clinical perspective.

Although hyperlipidemia (VLDL) may play a part of role in the onset of LA remodeling and AF, what is generally interpreted is, triggered by myocardial stretch of LA, the RAA system in atrial tissue is activated and the fibrosis of LA is promoted (structural remodeling). This, in turn, causes myocardial conduction non-uniformity, giving the anatomical base on which AF is maintained. Once AF develops, its conduction delays over time and finally becomes persistent AF (electrical remodeling).

Reply: We would like to thank the reviewer for pointing out the significance of the work and for the expert comments. In the following we respond to these comments.

Therefore, I think that the following points need to be added.

Atrial fibrosis has attracted attention as a cause of AF. It has been reported that atrial fibrosis progresses with aging, hypertension (HT), diabetes (DM), etc., and becomes a risk of developing AF. HT is closely related to LA remodeling and AF (about 30% of AF patients are patients with HT). In addition, in patients with MetS and hyperlipidemia often have HT and DM at the same time, thus clinically, it is often difficult to identify the cause of atrial dysfunction. Therefore, careful discussion is required.

Response:

We agree that clinically, it is often difficult to identify the cause of atrial dysfunction due to the co-existing HT and DM for patients with MetS and dyslipidemia. Thank you for this comment. Accordingly, we added some statements to address the existing knowledge regarding the role of HT and DM on atrial remodeling and fibrosis. Please see Line 98- 119.

What is the definition of atrial cardiomyopathy in the schematic diagram of Fig. 1? Does it mean a decrease in LA contractility? Active contraction (booster pump function) of LA disappears when AF develops.

Response:

Regarding the definition (classification) of atrial cardiomyopathy in the schematic diagram of Fig. 1, it is EHRAS Class IVf. The related description had been added to Section 3 (line 238-240) and the Figure 2 legend (line 658-659). For mice, we did not measure the LA contractility, which is an important parameter for LA function and we will try to do the measurement when we perform the next experiment.   

It is appropriate to understand clinically from the viewpoint that electrical remodeling and structural remodeling occur in parallel, and that AF is fixed and eventually develops into atrial cardiomyopathy.

   Therefore, Atrial cardiomyopathy→Atrial fibrillation is skeptical.

Response:

We would like to thank the reviewer’s comment. Some description to clarify the latest definition of atrial cardiomyopathy was added to Section 1.2 (line 60-64).

The conjoint group of EHRA/HRS/APHRS/SOLAECE published the expert consensus for the working definition of atrial cardiomyopathy: “Any complex of structural, architectural, contractile or electrophysiological changes affecting the atria with the potential to produce clinically-relevant manifestations”. In this consensus report, some description regarding the contribution of metabolic syndrome to atrial cardiomyopathy as following: “many diseases (like hypertension, heart failure, diabetes, and myocarditis) or conditions (like ageing and endocrine abnormalities) are known to induce or contribute to atrial cardiomyopathy. However, the induced changes are not necessarily disease-specific and pathological changes often share many similarities. The extent of pathological changes may vary over time and atrial location, causing substantial intra-individual and inter-individual differences.” With this new definition, atrial cardiomyopathy can precede the onset of AF by a couple of decades.

There are some cases showing AF reverse remodeling after ablation therapy in AF patients. How does author explain this phenomenon from viewpoint of MetS-related cardiomyopathy histologically?

Response:

Thank you for this comment. AF per se-induced atrial remodeling and the reverse remodeling after ablation therapy in AF patients is a big issue (no matter on structure or electricity) to address, and the topic can be another review paper. Regarding the differences between AF-induced remodeling (or reverse remodeling after a successful ablation) and MetS-induced cardiomyopathy, we added some conceptual description to Section 1.2 (line 78-80).

Is the pivotal role in line 33, the most important in line 322, etc. overemphasized?

Response:

The pivotal role for VLDL in the MetS-related atrial cardiomyopathy with vulnerability to AF is proposed based on our available research findings and existing knowledge. We had modified the statement regarding “the pivotal role” in the text (line 33-36 & line 316-317).

To address clinical relevance, we also added: “Further human studies are required to focus on the effects/correlation of postprandial lipids on atrial remodeling and its progression to determine whether VLDL-targeted therapy can reduce the occurrence of MetS-related AF”. (line 345-349).

Round 3

Reviewer 1 Report

The text has been corrected appropriately.